# Wharton’s Jelly Tissue Allograft for Connective Tissue Defects Surrounding Nerves in the Tarsal Tunnel: A Retrospective Case Series

**DOI:** 10.3390/reports7010008

**Published:** 2024-01-30

**Authors:** Ronald Bruton, Tracie L. Gilliland, John J. Shou, Crislyn G. Woods, Naomi E. Lambert, Tyler C. Barrett

**Affiliations:** 1Advanced Medicine of the Ozarks, Mountain Home, AR 72653, USA; drbruton@yahoo.com (R.B.); relief@amozarks.com (T.L.G.); 2Baylor College of Medicine, Houston, TX 77030, USA; drshou@regenativelabs.com; 3Regenative Labs, Pensacola, FL 32501, USA; crislyn@regenativelabs.com (C.G.W.); tyler@regenativelabs.com (T.C.B.)

**Keywords:** tarsal tunnel, nerve damage, neuropathy, Wharton’s jelly, regenerative medicine

## Abstract

Caused by age or trauma, collapsed connective tissue can cause nerve entrapment and damage within the tarsal tunnel. Tarsal tunnel syndrome is relatively underdiagnosed. This study presents an intervention targeting damaged tissues surrounding the nerves and replacing the structural cushioning with a Wharton’s jelly tissue allograft. The eight patients in our study, selected from four clinical sites, had tarsal tunnel-related defects. Patient outcomes were tracked on a 90-day calendar utilizing the Numeric Pain Rating Scale (NPRS) and the Western Ontario and McMaster University Arthritis Index (WOMAC). All patients had failed standard care practices for at least six weeks. Each patient received a Wharton’s jelly tissue allograft to sites around the affected tarsal tunnel. No patients experienced adverse reactions. The percent change results calculated from the initial application to the 90-day follow-up showed an improvement of 59.43% in NPRS and a 37.58% improvement in WOMAC. This study provides evidence that WJ allograft applications are safe, minimally invasive, and efficacious for patients who have failed standard care treatments for tissue defects associated with tarsal tunnel syndrome. The limitations of this study include its small cohort size and nonblinded nature. The results of this study warrant further research to confirm the efficacy, optimal dose, protocol, and durability of Wharton’s jelly.

## 1. Introduction

Tarsal tunnel syndrome is an entrapment neuropathy of the posterior tibial nerve and, potentially, its terminal branches under the flexor retinaculum and behind the medial malleolus of the ankle [1]. Tarsal tunnel syndrome can be characterized by local tenderness, pain, paresthesia, and heat, followed by numbness and tingling. Symptoms may become more permanent and severe, spreading toward the posterior, medial, or distal aspects of the lower extremity [1]. 

The occurrence of nerve damage in the tarsal tunnel is unclear and thought to be underdiagnosed. However, it has been found to have a higher incidence in females and can be witnessed at any age [2]. Contributing factors to the incidence of tarsal tunnel syndrome include trauma, tight-fitting shoes, abnormal biomechanics, and systemic diseases which may induce nerve or surrounding tissue inflammation [3]. When left untreated, posterior tibial nerve compression can cause permanent nerve damage, atrophy, and persistent pain [2]. 

While many intervention strategies exist for treating tarsal tunnel syndrome, there is limited robust evidence to guide the clinical management of the syndrome [4]. Currently, standard conservative management includes activity modification, physical rehabilitation, corticosteroid injections, and non-steroidal anti-inflammatory drugs (NSAIDs) [5]. It remains unclear when to intervene with surgical procedures as opposed to conservative management due to the various stages of the disease and the need for a structured, stepwise approach when treating patients [4]. Although the initiation of surgical intervention is unclear, there are a few different surgical approaches that are available. Three methods for decompression of the tibial nerve and its branches include open surgery, endoscopic surgery, and ultrasound-guided surgery [6]. Any surgical tarsal tunnel intervention can range in out-of-pocket costs from 3000 to 7000 USD, and patient recovery time can last up to six weeks. Novel alternative interventions are necessary for refractory connective tissue nerve damage, as surgery does not guarantee improvement, given that surgical success rates vary from 44% to 96% [5]. These interventions can require much of a patient’s time to be dedicated to the recovery process without a guarantee of lasting success. With uncertainty surrounding the treatment of tarsal tunnel defects, this study aims to propose an alternative intervention that targets explicit damage to the connective tissues surrounding the related nerves for patients who have failed conservative management and wish to avoid surgical intervention. 

Cases of nerve compression can have varying etiologies, but most equate to breakdown, rearrangement from trauma, and damage to the surrounding structural tissues. This collapse of connective tissue puts excess pressure on specific points along a given nerve, deteriorating the protective cushioning surrounding nerve fascicles. Most connective tissues, including the nerve extracellular matrix (ECM), comprise collagenic matrices as their primary structural support component [7]. Wharton’s jelly is a loose connective tissue found in the umbilical cord that cushions and protects the vessels within the cord from external forces and stretching. It contains collagen types I and III, hyaluronic acid, proteoglycans, growth factors, and cytokines. Hydrodissection of a compressed nerve with Wharton’s jelly can supplement the damaged protective coating and provide additional cushioning to the nerve, as well as replace surrounding collapsed connective tissues, promoting proper function. 

The retrospective repository used in this study, facilitated by Regenative Labs, contains data on over 180+ beneficial homologous uses for Wharton’s jelly tissue allografts, including musculoskeletal defects. This case series presents data from patient-reported pain scales in the retrospective repository of eight patients who received one application of Wharton’s jelly to refractory nerve damage and compression within the tarsal tunnel. This study serves to fill the literature gap regarding the use of Wharton’s Jelly for degenerative tissue defects. The purpose of this study was to examine and present the effect of Wharton’s Jelly application on supplementing structural defects in the connective tissue surrounding nerve damage and compression within the tarsal tunnel.

## 2. Detailed Case Description

### 2.1. Materials and Methods

All methods complied with the FDA and American Association of Tissue Banks (AATB) standards. This study was conducted under an Institute of Regenerative and Cellular Medicine IRB-approved protocol (RL-UCT-001), and informed consent was obtained from the study participants. Human umbilical cords were obtained from consenting donors following full-term cesarean section deliveries. Prior to delivery, donors underwent comprehensive medical, social, and blood testing. Qualtex Laboratories in San Antonio, TX, USA, tested all donations for infectious disease in accordance with the Clinical Laboratory Improvement Amendments (CLIA) of 1988, 42 CFR part 493, and FDA regulations. Each donor was tested for Hepatitis B Core Antibody (HBcAb), Hepatitis B, Surface Antigen (HBsAg), Hepatitis C Antibody (HCV), Human Immunodeficiency Virus Antibody, (HIV1/HIV-2 Plus O), Human T-Lymphotropic Virus Antibody (HLTV-I/11), Syphilis (RPR), Cytomegalovirus (CMV), HIV1/HCV/HBV, NAT, and West Nile Virus (WNV). Each test was performed using an FDA-approved testing kit. All test results were negative or non-reactive. All procedures were performed in accordance with strict aseptic techniques. In an ISO class 5 biologic safety cabinet, the umbilical cord was rinsed with saline to remove excess blood residue and clots. Wharton’s jelly was aseptically dissociated from the rinsed umbilical cord. After dissociation, 150 mg of Wharton’s Jelly was suspended in approximately 2 mL of sterile Sodium Chloride 0.9% solution (normal saline). The sample was not combined with cells, tissues, or articles other than the exceptions outlined in 21 CFR Part 1271.10(a) (Human Cells, Tissues, and Cellular and Tissue-Based Product Regulation). The manufacture of the HCT/P does not involve the combination of the cells or tissues with another article, except for water, crystalloids, or a sterilizing, preserving, or storage agent, provided that the addition of water, crystalloids, or the sterilizing, preserving, or storage agent does not raise new clinical safety concerns with respect to the HCT/P. Because the WJ has been tested for infectious diseases and it is not combined with another article except for water, crystalloids, and preservation agents, it has a very minimal risk of it inducing negative reactions. Likely due to the minimal risk of negative reactions, there have been no reported negative side effects or contraindications of WJ. Wharton’s jelly tissue allograft was then distributed by Regenative Labs.

### 2.2. Case Presentation

This retrospective case study pulled patients from the Regenative Labs repository with the following inclusion criteria: documented tarsal tunnel nerve-related defects who received only one 2 mL application of the 150 mg Wharton’s jelly tissue allograft and had complete data sets. Complete data sets are defined as having pain scales recorded at initial, 30-day, and 90-day visits with less than four blanks. Following these requirements, eight patients with nerve damage on one or both legs were identified from four clinics that submitted data. The contributing clinics included Enhanced Healthcare of the Ozarks, Baycity Associates in Podiatry, Regenerative Health 360, and Advanced Medicine of the Ozarks. The data sets were completed for each extremity separately. The pain scales utilized were the Numeric Pain Rating Scale (NPRS) and the function rating scale, as well as the Western Ontario and McMaster University Arthritis Index (WOMAC) (Figure 1) [8,9]. The severity of neuropathy among the participants in this study was determined at each clinic through several tests that assess the different nerve senses.

These tests included the Graphesthesia test, the Rebuilder Mitt Test, reflex reactions, the Romberg test, and the Tandem test. The Graphesthesia test interprets ambiguous tactile symbols from different spatial perspectives [10]. The Rebuilder Mitt test uses medical device mitts that send electrical signals that are exacerbated sensations of typical nerve signals to the nerves and muscles. The signals from the mitts serve to stimulate the nerves and strengthen the muscles. Reflex reactions were tested in the ankle as neuropathy affects sensory and motor components [11]. The Romberg test removes the visual and vestibular components that contribute to balance to identify a particular impairment in patients with proprioception difficulties [12]. The Tandem test is used to screen for neurological and vestibular disorders by having the patient close their eyes and walk. While the patient walks, the test administrator counts the number of consecutive tandem steps out of ten [13]. Additional techniques and devices to assess patient neuropathy include cold sensitivity, a pinwheel device to evaluate the subject’s ability to feel sharp or pointed sensations, vibrations through the use of a tuning fork, a medi-tip to test pinprick sensation and determine between sharp and dull sensations, and 10 g monofilament [14]. Temperatures in the feet, forearms, and face, along with oxygen in the feet and hands, were compared to assure symmetrical sensation. The purpose of these tests is to provide a baseline of sensory loss. If the results of the sensory test show that sensory loss has only occurred in the feet, then a specific amount of Wharton’s jelly is applied in specific anatomical sites of the foot. 

This study included eight patients, one female and seven males, who presented with nerve defects in the tarsal tunnel aspect of their lower extremities. Two patients received WJ in their right foot only, and four patients received WJ in their left foot. Two patients received WJ in both their feet. The age distribution included one patient in the 40–49 range, six in the 70–79 range, and one in the 80–89 range. Regarding BMI distribution, four patients were categorically overweight, two patients were obese, and two patients had an unreported BMI. 

### 2.3. Procedure

A 25-gauge needle was used in the application of the WJ tissue allograft. The application did not involve a guided entry. If sensory loss was present only in the foot, a total of 2 mL WJ was applied in 3 separate injection sites. The sites included 0.5 cc of WJ at the posterior tibial nerve, 0.5 cc at the medial plantar nerve, and 1 cc at the superficial peroneal nerve on the dorsal side of the foot. If the neuropathy extended upwards towards the patella, then a total of 2 mL WJ was applied in four different injection sites: 0.5 cc into the lateral calcaneus branch, 0.5 cc to the lateral peroneal nerve just below the patella, 0.5 cc to the medial plantar nerve, and 0.5 cc to the posterior tibial nerve (Figure 1 and Figure 2).

### 2.4. Results

The percent changes regarding the improvement in patient pain scales were calculated using the cohort averages at initial application, 30-day follow-up, and 90-day follow-up. The average NPRS score was 6.7 upon initial application, and the average WOMAC at this point was 30.6. At the 30-day follow-up, NPRS was 4.8, and WOMAC was 26.8. At the 90-day follow-up, NPRS was 2.75, and WOMAC was 19.1. The standard deviation of each mean is represented in Table 1. The large deviation in WOMAC averages is due to some patients not maintaining any feeling in their lower extremities, while others reported physical pain with some numbness before the application. Individual patients’ total score for each scale at the data collection dates can be found in Appendix A. Percent improvement was calculated for NPRS and WOMAC from initial appointment to the 30-day and 90-day follow-up appointments. From the initial application to the 30-day follow-up, there was a 41.20% improvement in NPRS and a 14.18% improvement in WOMAC. Finally, from initial application to 90-day follow-up, an improvement of 59.43% in NPRS and a 37.58% improvement in WOMAC were achieved. Overall, the most significant improvement was seen in the NPRS category from initial application to the 90-day follow-up, but all patients experienced significant improvements in pain. Figure 2 compares the percent improvement in the NPRS and WOMAC scales. The figure illustrates a steep improvement in NPRS from initial application to 30 days after application and a gradual improvement from 30 to 90 days. In comparison, the WOMAC experienced a steady, gradual improvement from application to 90-day follow-up. Figure 3 illustrates individual tarsal tunnel data sets for the WOMAC scale. It is essential to recognize that a higher WOMAC score correlates to increased pain. 

## 3. Discussion

Given the pain improvements reported on various pain rating scales, this study provides evidence that WJ allograft applications are safe, minimally invasive, and efficacious for patients who have failed standard care treatments for connective tissue defects associated with tarsal tunnel syndrome. Of the patients in this study, no adverse reactions were reported. Wharton’s jelly is extracted from a human umbilical cord, classifying it as an immune-privileged tissue, meaning it will not elicit an immune reaction. Although the umbilical cord tissue has been thoroughly tested for infectious diseases, minimal risk remains. It should also be noted that the tissue is cryopreserved and protected with DMSO. If the patient receiving the tissue allograft has an allergy to sulfonamides, there is a DMSO-free allograft available. Provided that a physician applies the product, human error can exist in the application process, which could potentially lead to an adverse reaction from injection site irritation. The results of this study warrant further research to confirm the efficacy of Wharton’s jelly and the potential for it to be added to conservative care protocols. Additional studies may clarify the optimal dose, protocol, and durability of WJ allograft application. 

The limitations of this study include its small cohort size and non-blinded trial design. This study’s small cohort size may have led to less precise results. In addition, it may not accurately represent how the product would work more generally. In future studies, an effort will be made to attain a larger sample size to produce more reliable and statistically analyzable results. The effect of the survey being non-blinded was minimized by using patient-reported scales (NPRS and WOMAC) which quantize patient pain, functionality, and stiffness based on an array of questions based on the patient’s perception of their own pain. Additionally, this study was retrospective in nature, which limited the use of site-specific scales and ultimately limited data collection, affecting the specificity of our results. 

The positive results presented in this retrospective case series align with the current literature on human tissue defects associated with knee pain and articular cartilage defects affiliated with the sacroiliac joint [15,16]. Patient-reported pain, joint stiffness, and physical function had a significant improvement in the knee study by Timmons. This study showed lasting pain relief maintained for more than 24 months after just one WJ application. The patient’s pain and progress were determined through the use of the Visual Analogue Scale (VAS) and WOMAC scale. The study included 30 adults with an average age of 63 years old. There was a statistically significant improvement in VAS scores with activity and with rest. Figure 4 shows the mean VAS scores at rest and with activity over time, along with the standard deviation. Similarly, the Womac scores improved from baseline over time with a *p*-value less than 0.001. Figure 5 shows the decrease in WOMAC scores over time. Overall, the study showed an improvement in NPRS and WOMAC scores, along with a reduction in opiate and NSAID use. In the sacroiliac study by Lai, 84% of the patients reported a reduction in NPRS, and 76% of the patients reported a reduction in WOMAC scores. This study analyzed a total of 38 patients with a mean age of 71 years. Overall, the percent change analysis showed an average improvement of 42% in NPRS scores and an average improvement of 22% in WOMAC scores at the 90-day mark. Table 2 shows the statistical significance of the WOMAC scores from the initial appointment and the final average scores. In these studies, no adverse reactions were reported, and significant pain improvement was seen in each study, aligning with the positive results in the presented case series, confirming WJ allografts as a promising alternative intervention for musculoskeletal and tissue defects. 

This study highlights the significant potential of WJ to aid in the tissue repair of collapsed connective tissues surrounding nerves and potentially the nerve ECM as it demonstrates improved nerve sensation among patients enduring neuropathy associated with tarsal tunnel syndrome. WJ can be utilized homologously in patients suffering from neuropathy when directly addressing the connective tissues surrounding individual nerve fibers in three distinct components: the endoneurium, perineurium, and epineurium [17]. The individual nerve fiber is surrounded by the endoneurium, which is composed of a network of collagen fibrils that serve to hold together the nerve fibers and blood capillaries in larger nerve fiber bundles. The perineurium, followed by the epineurium, surrounds the endoneurium. The endoneurium, perineurium, and epineurium are all connective tissues with individual responsibilities that serve together as shielding and cushioning barriers for the impulse-conducting elements of the myelin sheath covering the nerve [18]. Specifically important to this study is the function of the epineurium. 

The epineurium primarily comprises a collagenous extracellular matrix surrounding the entire nerve, contributing to nerve tensile strength [18]. Nerve damage leading to neuropathy may occur if the tissue surrounding the nerve does not fully support the nerve. When tissues endure pressure, they deform and create pressure gradients. Often, compression occurs at sites where a nerve runs through a tunnel that is formed by stiff tissue boundaries [19]. A study by Rempel tested the effects of compression damage on nerve sensation and found that axonal degeneration occurred with compression and correlated with endoneurial edema [19]. A study by Gao explains that the ECM begins to participate in the nerve regeneration process in that the epineurium, perineurium, and endoneurium, which are composed of collagen, provide structural support for nerve regeneration [7]. Like the ECM of the nerve fiber, the primary function of WJ is to provide cushion, protection, and structural support to umbilical vessels by preventing their compression, torsion, and bending [20]. This study proposes that the application of WJ to the surrounding area of the affected nerve can supplement and promote the repair of the damaged connective tissue that is contributing to nerve compression. When WJ is applied directly to the nerve, it can replace the missing ECM and provide cushioning and support to the nerve fascicles, promoting standard functionality. This homologous supplementation is supported by the positive outcomes reported in this study.

Given the function and components of WJ and the significant results in this study, WJ presents a promising alternative to the current standard-of-care practices and could potentially prevent further invasive procedures. Additional studies stemming from randomized control trials could statistically compare the efficacy and durability of WJ in connective nerve tissue supplementation for neuropathy patients with other standard non-invasive procedures.

## 4. Conclusions

The utilization of WJ allografts in supplementing tissue defects associated with tarsal tunnel syndrome leads to improvement in patient pain and function. WJ can replace the damaged ECM and connective layers of the affected nerves and cushion the nerve from exterior soft tissue damage, which leads to improved nerve sensation, ultimately decreasing neuropathy associated with tarsal tunnel syndrome. After failing other standard-of-care treatment options, the patients in this study were able to find relief via a single WJ allograft application. These findings suggest that WJ allografts can be used as an effective intervention for defects related to tarsal tunnel syndrome when the standard-of-care treatments have failed. The utilization of WJ allografts could decrease the occurrence of surgical procedures and ultimately be more time-consuming and cost-effective. Future studies may include a larger and more diverse cohort and a blinded control group to evaluate the safety and efficacy of WJ further and assist in defining dosage protocols in the application of tissue defects associated with tarsal tunnel syndrome. 

## Data Availability

Data can be found in Appendix A (Figure A1).

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
