# Peer review of "Wharton’s Jelly Tissue Allograft for Connective Tissue Defects Surrounding Nerves in the Tarsal Tunnel: A Retrospective Case Series"

_reports, 2024, doi:10.3390/reports7010008_

Round 1

Reviewer 1 Report

Comments and Suggestions for Authors

The manuscript discusses the use of Wharton's jelly tissue allograft for connective tissue defects surrounding nerves in the tarsal tunnel. This case series presents a novel approach using Wharton's jelly tissue allograft to replace the damaged connective tissue and provide cushioning and support to the affected nerves. The study is well-designed; however, the manuscript could benefit from some improvements. Here are some suggestions to enhance the overall quality of the manuscript.

1. In the abstract, it would be helpful to mention the limitations of the study, including the small cohort size and non-blinded trial design.

2. In the "Introduction" section, when discussing the current standard conservative management and surgical interventions for tarsal tunnel syndrome, it would be valuable to mention any limitations or challenges associated with these approaches. This will provide a more comprehensive view of the treatment landscape and highlight the need for alternative interventions.

3. In the subsection "2.1.1. Case Presentation," it would be beneficial to provide a concise explanation of the Numeric Pain Rating Scale (NPRS) to ensure clarity for readers who may not be familiar with this specific pain scale. Additionally, it is important to include the relevant citation for the method used appropriately.

4. In the subsection "2.2. Results," consider providing the standard deviation or range of the NPRS and Western Ontario and McMaster University Arthritis Index (WOMAC) scores at each time point to give readers a sense of the variability in the data.

5. Figure 1 should be accompanied by a descriptive and informative caption to provide context and enhance the understanding of the depicted information.

6. In the "Discussion" section, it would be beneficial to discuss the potential mechanisms by which Wharton's jelly (WJ) allografts may contribute to pain management and tissue repair. Exploring the biological basis or hypothesized mechanisms of action can add depth to the discussion and support the study's findings.

7. Consider discussing the limitations of the study in more detail. While the small cohort size and non-blinded trial design are mentioned, it would be valuable to elaborate on the potential impact of these limitations on the study's conclusions and the generalizability of the results.

8. When discussing the potential of WJ allografts, it would be beneficial to address any known risks or potential complications associated with this intervention. While the study reports no adverse reactions, it's important to acknowledge that adverse events can occur and discuss their potential implications.

9. In the "Conclusions" section, it would be helpful to mention the potential impact of this research on clinical practice. Discussing how the findings may influence treatment approaches or offer an alternative intervention for patients who have failed standard-of-care treatments would add practical relevance to the conclusions.

Author Response

We thank you greatly for the time you took to review our manuscript and send in your helpful edits. We have made changes to the manuscript accordingly and present our responses to your comments below. 

  1. A few limitations of the study were added to the abstract. The limitations mentioned include a small cohort size and a non-blinded trial design.
  2. We clarified the limitations of the current standard of care approaches
  3. We included an image of the pain scales for reference and added a reference for each scale. 
  4. We added a table with the mean and standard deviation for each data collection point and then discussed the points. 
  5.  We added descriptive captions to all figures.
  6.  The second to last paragraph of the discussion describes our hypothesized mechanism of connective tissue supplement with WJ to compressed nerves surrounded by collapsed tissue.
  7.  While there was a sentence in the discussion section briefly addressing the limitations of the study, the sentence was separated into its own paragraph. Additionally, sentences were included to further explain the limitations of the study and their effects. 
  8.  In the first paragraph of the discussion section, product risks have been discussed along with precautionary measures. 
  9. A couple of sentences have been added toward the end of the conclusion section to discuss ways in which WJ allografts have a potential impact. 

Reviewer 2 Report

Comments and Suggestions for Authors

In this manuscript, the authors reported a new method to treat  tarsal tunnel syndrome  with a Wharton’s jelly tissue allograft. The results showed that it may be a safe, minimally invasive, and efficacious for patients who have failed standard care treatments for Tarsal Tunnel syndrome. It's an interesting study and provides some new ideas to the readers. However, there're still several issues which should be addressed.

1. Some patients received high-powered laser therapy or red-light therapy as well as vibration therapy. These could cause confounding bias. Please provide the detailed information. 

2. How did the  producer get Wharton's jelly? Are there any ethical risks? What techniques are used to treat Wharton's jelly before clinical use? How to decellularize? 

3. Are there contraindications of this treatment? Are there any other risks?

4. A paragraph should be added to the section of  discussion to explain the limitations of this treatment.

So, minor revision should recommended. 

Author Response

We thank you greatly for the time you took to review our manuscript and send in your helpful edits. We have made changes to the manuscript accordingly and present our responses to your comments below.

  1. In an effort to address the confounding bias of Some patients receiving high-powered laser therapy or red-light therapy as well as vibration therapy, the sentence has been deleted in its entirety. We decided to delete the sentence as it was simply a statement of what we have seen multiple providers perform throughout various application sites. Thus, it was merely a suggestion of techniques that could be performed in conjunction with the tissue allograft. However, the statement did not reflect the specific procedure in regard to Tarsal Tunnel tissue defects, so the statement was removed. 
  2. All questions regarding the production of WJ have been answered in the first paragraph under the Materials and Methods section. 
  3. There have been no reported contraindications to the use of WJ, and the associated risks of the product have been discussed towards the end of the materials and methods section. 
  4. While there was a sentence in the discussion section briefly addressing the limitations of the study, the sentence was separated into its own paragraph. Additionally sentences were included to further explain the limitations of the study and their effects. 

Reviewer 3 Report

Comments and Suggestions for Authors

The scientific paper "Wharton’s Jelly Tissue Allograft for Connective Tissue Defects Surrounding Nerves in the Tarsal Tunnel: A Retrospective Case Series” highlights a novel approach (retrospective case series), targeting the damaged tissues surrounding the nerves and replacing the structural cushioning with a Wharton’s jelly tissue allograft.

It can be considered that:

1)      Reduce the abstarct to 250 words. I suggest reducing the background.

2)      Author affiliations are incomplete.

3)      In the abstract, specify whether the red light is laser or LED.

4)      The citation of references (called in the text) are outside the periodical standard.

5)      The names of the authors of the manuscript are not in the periodical standard.

6)      After the references, all the elements requested by the journal are missing: conflict of interest, contributions from each author, ethical aspects, among others. Please pay attention to the instructions to the journal's authors.

7)      The introduction is written in 2 paragraphs. I recommend adjusting it to 5 paragraphs, with the last paragraph explaining the gap in the literature and the objective of the study.

8)      The authors need to explain and highlight neuropraxia injuries (use this term).

9)      Specify the ethics committee that approved the study, protocol number and approval date.

10)  The methodology is poorly detailed and the number of patients does not allow us to reach conclusions. They should be clearer: inclusion and exclusion criteria, how many patients were initially recruited and how the number of 8 patients was arrived at.

11)  The description of photobiomodulation therapy is completely incomplete. What was the protocol? Describe it completely, as this totally interferes with the rehabilitation process (pain, edema, function, etc.).

12)  The results are poor. It could present pain quantification data, for example. More data collection could have been done that are of interest for the type of injury.

13)  The discussion has little comparison with previous studies. It can be observed by the small number of references used in this section and in this study.

14)  Include study limitations in the discussion.

Comments on the Quality of English Language

Moderate editing

Author Response

We thank you greatly for the time you took to review our manuscript and send in your helpful edits. We have made changes to the manuscript accordingly and present our responses to your comments below. 

  1. The abstract has been reduced to 200 words per MDPI guidelines. 
  2. The affiliations have been completed according to the MDPI guidelines
  3. In order to reduce the amount of words in the abstract, the statement about red lights has been removed. This statement was removed as it was not significant to the study. Red light therapy had only been mentioned as a recommendation due to several physicians using it in conjunction with WJ allograft. 
  4. The citations have been edited according to the MDPI Guidelines 
  5. The author names have been adjusted according to the MDPI Guidelines
  6. We added the required sections according to the MDPI guidelines
  7. The introduction has been divided into five paragraphs. In the last paragraph of the introduction, the literature gap and objective of the study have been stated. 
  8. Neuropraxia does not affect the connective tissue surrounding the nerve, which is what the WJ is used to supplement. While the patients might also have neuropraxia, the WJ cannot repair the actual nerve but rather the surrounding tissues to cushion it from external forces. 
  9. The specific IRB committee and approval number were added to the institutional review board statement at the end of the paper
  10. We more clearly described the inclusion criteria we followed when identifying the patient data set to be used from the retrospective repository. 
  11. We have decided to remove the discussion of laser therapy completely. Laser therapy had only been discussed in the study as some physicians use it as their own preference externally from the main protocol for the repository. We initially included it to bring to light how different physicians use the product and what additional techniques they perform. We decided to remove laser therapy from the report since it was not used in the study protocol. Thus, the effects of laser therapy had not been independently studied. 
  12. Because this is a retrospective study, there is a limitation to what scales were used and what data was collected. The NPRS and WOMAC scale functions to quantify patient-reported pain for lower extremities in general, but future studies could incorporate more specific scales. 
  13. Few comparisons were made as there is a large gap in the literature regarding the use of Wharton’s Jelly. While we have several published articles regarding the use of WJ in multiple application sites, they were not used for comparison in an effort to avoid referencing ourselves. However, to provide more comparison, more information from the original referenced sources has been discussed. Additionally, graphs from the studies have been added to add value further. 
  14. While there was a sentence in the discussion section briefly addressing the limitations of the study, the sentence was separated into its own paragraph. Additionally sentences were included to further explain the limitations of the study and their effects. 

Reviewer 4 Report

Comments and Suggestions for Authors

The value of the paper will increase if you add some clinical pictures during procedure with the technique steps. Also, in the discussion section it would be useful to add a table containing other studies and their results in the field, with the same topic.

Comments on the Quality of English Language

Minor corrections are needed

Author Response

We thank you greatly for the time you took to review our manuscript and send in your helpful edits. We have made changes to the manuscript accordingly and present our responses to your comments below. 

We added some diagrams depicting the application sites and volumes that are described in the procedure section to clarify the Wharton’s jelly applications further. 

A table from the referenced sources and additional findings have been added to the discussion section to add depth to the comparisons. The referenced sources both address the use of Wharton's Jelly in the same use as it was utilized in this study. However, it was applied in different application sites. A source utilizing WJ for degenerative defects of the tarsal tunnel has not been sourced as, to our knowledge, we are the first to write about the use of WJ connective tissue for tarsal tunnel defects.

Round 2

Reviewer 1 Report

Comments and Suggestions for Authors

I appreciate the effort the authors have put into revising the paper, which has significantly enhanced its quality. Based on the revisions made, I recommend accepting the paper for publication.

Reviewer 3 Report

Comments and Suggestions for Authors

No comments

Comments on the Quality of English Language

Minor editing